# Fast Projection onto the Capped Simplex with Applications to Sparse Regression in Bioinformatics

**Andersen Ang**[*]
Dept. of Combinatorics and Optimization
University of Waterloo
ms3ang@uwaterloo.ca

**Jianzhu Ma**
Institute for Artificial Intelligence
Peking University
majianzhu@pku.edu.cn

**Nianjun Liu**
Dept. of Epidemiology and Biostatistics
Indiana University Bloomington
liunian@indiana.edu

**Kun Huang**
Dept. of Biostatistics and Health Data Science
Indiana University
kunhuang@iu.edu

**Yijie Wang**[*]
Dept. of Computer Science
Indiana University Bloomington
yijwang@iu.edu

## Abstract

We consider the problem of projecting a vector onto the so-called k-capped simplex, which is a hyper-cube cut by a hyperplane. For an n-dimensional input vector with bounded elements, we found that a simple algorithm based on Newton's method is able to solve the projection problem to high precision with a complexity roughly about O(n), which has a much lower computational cost compared with the existing sorting-based methods proposed in the literature. We provide a theory for partial explanation and justification of the method.

We demonstrate that the proposed algorithm can produce a solution of the projection problem with high precision on large scale datasets, and the algorithm is able to significantly outperform the state-of-the-art methods in terms of runtime (about 6-8 times faster than a commercial software with respect to CPU time for input vector with 1 million variables or more).

We further illustrate the effectiveness of the proposed algorithm on solving sparse regression in a bioinformatics problem. Empirical results on the GWAS dataset (with 1,500,000 single-nucleotide polymorphisms) show that, when using the proposed method to accelerate the Projected Quasi-Newton (PQN) method, the accelerated PQN algorithm is able to handle huge-scale regression problem and it is more efficient (about 3-6 times faster) than the current state-of-the-art methods.

## 1 Introduction

The $k$-capped simplex [17] is defined as $\Delta_k := \left\{ \mathbf{x} \in \mathbb{R}^n \,\middle|\, \mathbf{x}^\top \mathbf{1} \leq k, \, \mathbf{0} \leq \mathbf{x} \leq \mathbf{1} \right\}$, where $k \in \mathbb{R}$ is an input parameter, $\mathbf{0}$ and $\mathbf{1}$ are vectors of zeros and vector of ones in $\mathbb{R}^n$, respectively (resp.), and the sign $\leq$ is taken element-wise. Geometrically, for some $k > 0$, the set $\Delta_k$ represents an $n$-dimensional hyper-cube $[\mathbf{0}, \mathbf{1}]^n$ with a corner chopped off by the half-space $\mathbf{x}^\top \mathbf{1} \leq k$. If the half-space constraint $\mathbf{x}^\top \mathbf{1} \leq k$ in $\Delta_k$ is replaced by the hyper-plane constraint $\mathbf{x}^\top \mathbf{1} = k$, for some $k > 0$, this set represents the slice of the hyper-cube cut by the hyper-plane, and we denote this set as $\Delta_k^=$. In this paper, we consider the problem of projecting a vector $\mathbf{y}$ onto the $k$-capped simplex:

$$\mathbf{x}^* = \text{proj}_{\Delta_k}(\mathbf{y}) = \underset{\mathbf{x}}{\text{argmin}}\left\{ \frac{1}{2}\|\mathbf{x} - \mathbf{y}\|_2^2 \text{ s.t. } \mathbf{x} \in \Delta_k \text{ (or } \Delta_k^=) \right\}. \tag{1}$$

---

[*]Both authors contributed equally.

35th Conference on Neural Information Processing Systems (NeurIPS 2021).

**Remark 1.** (*On the input* $k$) *In this paper, we assume* $0 < k < \infty$ *is properly selected such that the set* $\Delta_k$ *(and* $\Delta_{\overline{k}}^{=}$ *) has a nonempty interior, and therefore the projection is feasible. I.e., we assume* $\left\{ \mathbf{p} \in \mathbb{R}^n \mid \mathbf{p}^\top \mathbf{1} \leq k \right\} \cap \left\{ \mathbf{q} \in \mathbb{R}^n \mid \mathbf{0} \leq \mathbf{q} \leq \mathbf{1} \right\} \neq \varnothing$ *(and similarly for* $\Delta_{\overline{k}}^{=}$ *). In this setting, a unique global minimizer* $\mathbf{x}^*$ *of Problem* (1) *exists, since the function* $\|\mathbf{x} - \mathbf{y}\|_2^2$ *is strongly convex. The value of* $k$ *is closely related to the root of a piecewise-linear equation that plays a critical role in the solution of Problem* (1)*, see discussion in* § 2*.*

Problem (1) generalizes the projection onto the probability simplex [4, 5] which has many applications. In this paper, we consider the application in sparse regression in bioinformatics, see § 4.

To the best of our knowledge, for an input vector $\mathbf{y} \in \mathbb{R}^n$, the state-of-the-art algorithm [17] for solving the Projection (1) has a time complexity of $\mathcal{O}(n^2)$. We found that a simple heuristic algorithm based on Newton's method is able to solve the same problem to high precision with a complexity roughly $\mathcal{O}(n)$ on average, based on exploiting the 2nd-order information of a scalar minimization problem on the Lagrangian multiplier of Problem (1); see § 2.3.

**Contributions**   We proposed a simple algorithm based on Newton's method, as an alternative to the sorting-based method to solve Problem (1). We first convert the problem into a scalar minimization problem, then we show that such scalar problem can be solved using Newton's iteration, see Theorem 1. We then show that the proposed method is superior to the current state-of-the-art method in terms of runtime. For a vector with dimension $10^6$, the proposed method is about 6-8 times faster than a commercial solver. Furthermore, we give numerical results to show the effectiveness of the proposed method on solving sparse regression in a bioinformatics problem on a real-world dataset with one million data points, see § 4.

**Scope, limitation, and impact**   We emphasize that the proposed method is not aimed at replacing all the well-established sorting-based methods in the literature, but to provide an alternative method for solving the projection problem in a special case. To be specific, we found that, when the input vector $\mathbf{y}$ has bounded elements (i.e., $|y_i| \leq \alpha$, for $\alpha$ up to 1000), the proposed algorithm in this paper is able to solve the projection problem much faster than the existing methods in the literature. It is important to note that, when the bound $\alpha$ becomes very huge (say a million), the proposed method sometimes converges slower than existing methods. Although limited in the scope of application, however, we argue that the proposed method is still very useful for applications that the input vectors are bounded. Using bioinformatics as an example, on an ordinary PC, we are able to reduce the runtime on identifying the genetic factors that contribute to dental caries (i.e., tooth decay) from half-hour to just 3 minutes; see § 4.

**Paper organization**   In § 2, we give the background material, show the main theorem, and discuss the proposed algorithm. In § 3, we show experimental results to verify the efficiency of the proposed algorithm on solving Problem (1). In § 4, we illustrate the effectiveness of the proposed method on solving sparse regression in a large bioinformatics dataset. We conclude the paper in § 6 and provide supplementary material in the appendix.

## 2   Projection onto the capped simplex

In this section, we discuss how to solve Problem (1). By introducing a Lagrangian multiplier $\gamma$, we convert (1) to a scalar problem in the form of $\min_\gamma \omega(\gamma)$, see (5). Then, we provide a theorem for characterizing the function $\omega(\gamma)$. More specifically, we show that $\omega(\gamma)$ is $n$-smooth, i.e., the Lipschitz constant of the derivative $\omega'(\gamma)$ grows linearly as the dimension $n$, which makes the gradient method ineffective to solve Problem (5) due to very small stepsize for input vector $\mathbf{y}$ that has many variables. This motivates us to switch to the 2nd-order method, i.e., Newton's method, to solve Problem (5). We end this section by discussing the properties of proposed Newton's method.

### 2.1   Reformulate the projection problem to a scalar minimization problem

Consider the case of projection onto $\Delta_{\overline{k}}^{=}$ [2]. Problem (1) is equivalent to Problem (2): by introducing a Lagrangian multiplier $\gamma$ that corresponds to the equality constraint, we form a partial Lagrangian $\mathcal{L}(\mathbf{x}, \gamma)$, and arrive at the min-max problem (2), where we keep the inequality constraint on $\mathbf{x}$ as

$$\min_{\mathbf{0} \leq \mathbf{x} \leq \mathbf{1}} \max_{\gamma \in \mathbb{R}} : \left\{ \mathcal{L}(\mathbf{x}, \gamma) = \frac{1}{2}\|\mathbf{x} - \mathbf{y}\|^2 + \gamma(\mathbf{x}^\top \mathbf{1} - k) \right\}. \tag{2}$$

---

[2] For projection onto $\Delta_k$, we can use the same approach by adding nonnegativity constraint $\gamma \geq 0$ in (2).

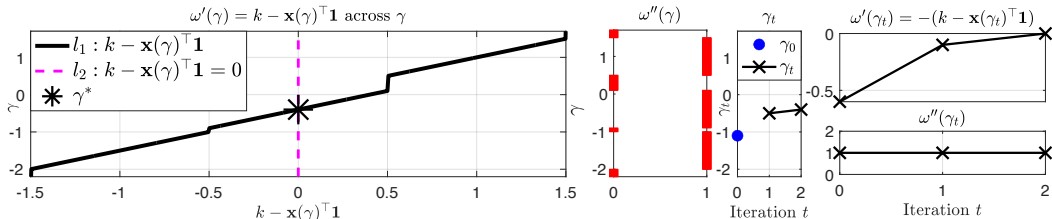

Figure 1: Illustration on solving Problem (5) on a toy example with $\mathbf{y} = [0.1, 1.5, -1] \in \mathbb{R}^3, k = 1.5$ and $\gamma_0 = -1.1$. The line $l_2$ intersects the piecewise-liner function $l_1$ at $\gamma^*$, which is the root of $l_2$. The idea of the sorting-based methods is to find such root by sorting the 3 elements of $\mathbf{y}$ and try each of the 3 linear segment of $l_1$ to find the root. The proposed method (Algorithm 1) solves the problem in 2 iterations, hence it gains speed-up. In the middle we plot $\omega''$ and the iteration of $\gamma_t$. The right-most plots show the iteration of $\omega'$ and $\omega''$ (see Theorem 1 for their explicit expression). Note that $l_1, l_2$ may not intersect each other for some $(\mathbf{y}, k)$, in this case $l_2$ has no root and the sorting-based methods do not work. However, the proposed algorithm can still produce an approximate solution.

**Closed-form solution on x**   Let $\gamma^*$ be the optimal $\gamma$ in (2), then the minimizer $\mathbf{x}^*$ of (2) is

$$\mathbf{x}^*(\gamma^*) = \underset{\mathbf{0} \leq \mathbf{x} \leq \mathbf{1}}{\operatorname{argmin}} \left\{ \mathcal{L}(\mathbf{x}) = \sum_{i=1}^n \frac{1}{2} x_i^2 + (\gamma^* - y_i) x_i \right\} \overset{(4)}{=} \min \left\{ \mathbf{1}, [\mathbf{v}(\gamma^*)]_+ \right\}, \qquad (3)$$

where $[\,\cdot\,]_+ = \max\{\,\cdot\,, 0\,\}$, both $\max, \min$ in (3) are taken element-wise, and $\mathbf{v}(\gamma)$ is defined as

$$\mathbf{v}(\gamma) = \mathbf{y} - \gamma \mathbf{1}. \qquad (4)$$

As stated in the introduction, under a sufficiently large $k > 0$, such $\mathbf{x}^*$ exists in the relative interior of $\Delta_{\overline{k}}^=$, which implies Slater's condition is true and thereby strong duality holds, i.e., we have the first equality sign in (3). Furthermore, we can swap the order of the min-max in (2) since it has a saddle point. By swapping the order of min-max, and using (3), we convert Problem (2) to a scalar minimization problem on $\gamma$:

$$\min_{\gamma \in \mathbb{R}} \; : \; \left\{ \omega(\gamma) := -\mathcal{L}\Big( \min \big\{ \mathbf{1}, [\mathbf{v}(\gamma)]_+ \big\}, \gamma \Big) \right\}, \qquad (5)$$

where the negative sign comes from flipping the maximization on $\gamma$ to minimization. Now, it is clear that if we solve (5), we can construct the solution $\mathbf{x}^*$ as in (3) to solve Problem (1). Many papers [4, 5, 17] in the literature follow such a direction on solving the projection problem, in which they solve Problem (5) using a sorting-based technique to find the root of the piecewise-linear function $\omega'(\gamma) = 0$ (see Theorem 1 for the expression of $\omega'$).

In this paper, we solve (5) by Newton's method. It is important to note that we can solve the same problem using gradient descent, but we found that it has a slow convergence, because $\omega(\gamma)$ can have a huge smoothness constant (see Theorem 1) and thus the feasible stepsize for the gradient step such that the function value will decrease is very small, which contributes to a slow convergence. We also observed that empirically the proposed method in this paper (see Algorithm 1) yields a much faster convergence (about 6-8 times faster) than the sorting-based methods, especially when the input vector $\mathbf{y}$ has a huge dimension; see § 3 for the experimental results.

Before we proceed, we illustrate the idea of solving the projection on a toy example, see Fig. 1.

## 2.2   The derivatives and the huge smoothness constant

Now we talk about the derivatives of $\omega(\gamma)$ with respect to (w.r.t.) $\gamma$. First, we show that $\omega(\cdot)$ is continuously differentiable and we derive its 1st and 2nd order derivatives. Then we show that the Lipschitz constant $L$ of $\omega'(\cdot)$ is $n$, which is the dimension of the input vector $\mathbf{y} \in \mathbb{R}^n$. With a huge $L$, we observed that gradient descent has a slow convergence on minimizing $\omega(\gamma)$, hence we propose to use the 2nd-order method, i.e., Newton's method, to solve (2); see § 2.3.

We now present the main theorem in this paper, which is about the derivatives of $\omega(\gamma)$.

**Theorem 1.** *The function $\omega(\gamma)$ is convex and twice differentiable with*

$$\omega'(\gamma) = k - \min\Big(\mathbf{1}, [\mathbf{v}(\gamma)]_+\Big)^\top \mathbf{1} \quad \text{is } n\text{-Lipschitz, and} \quad \omega''(\gamma) = \sum_{i=1}^n I_{0 < v_i < 1} \geq 0, \qquad (6)$$

*where $I$ is an indicator function that $I_A = 1$ if $A$ is true and $I_A = 0$ otherwise.*

Now we present two lemmas to prove the Theorem 1. For the first lemma, recall that for all $a, b \in \mathbb{R}$,

$$\max\{a, b\} = \frac{1}{2}\Big(a + b + |a - b|\Big) \quad \text{and} \quad \min\{a, b\} = \frac{1}{2}\Big(a + b - |a - b|\Big). \qquad (\star)$$

**Lemma 1.** $\forall \mathbf{x}, \mathbf{y} \in \mathbb{R}^n$, we have $\Big\| \min\big(\mathbf{1}, [\mathbf{x}]_+\big) - \min\big(\mathbf{1}, [\mathbf{y}]_+\big) \Big\|_2 \le \|\mathbf{x} - \mathbf{y}\|_2$.

*Proof.* By triangle inequality and $(\star)$, for all $a, b \in \mathbb{R}$, we have $\big| \min(a, 1) - \min(b, 1) \big| < \big| a - b \big|$ and $\big| [a]_+ - [b]_+ \big| < \big| a - b \big|$. Therefore,

$$\big\| \min(\mathbf{1}, [\mathbf{x}]_+) - \min(\mathbf{1}, [\mathbf{y}]_+) \big\|_2 \le \big\| [\mathbf{x}]_+ - [\mathbf{y}]_+ \big\|_2 \le \|\mathbf{x} - \mathbf{y}\|_2.$$

$\square$

The second lemma is related to the notion of optimal value function [3, Section 4].

**Lemma 2.** *[Theorem 4.1 [3]] Let $\mathcal{X}$ be a metric space and $\mathcal{U}$ be a normed space. Suppose that for all $\mathbf{x} \in \mathcal{X}$, the function $\mathcal{L}(\mathbf{x}, \cdot)$ is differentiable and that $\mathcal{L}(\mathbf{x}, \gamma)$ and $D_\gamma \mathcal{L}(\mathbf{x}, \gamma)$ (the partial derivative of $\mathcal{L}(\mathbf{x}, \gamma)$ w.r.t. $\gamma$) are continuous on $\mathcal{X} \times \mathcal{U}$. Let $\Phi$ be a compact subset of $\mathcal{X}$. Then the optimal value function $\phi(\gamma) := \inf_{\mathbf{x} \in \Phi} \mathcal{L}(\mathbf{x}, \gamma)$, is (Hadamard) directionally differentiable. In addition, if $\forall \gamma \in \mathcal{U}$, $\mathcal{L}(\cdot, \gamma)$ has a unique minimizer $\mathbf{x}(\gamma)$ over $\Phi$, then $\phi(\gamma)$ is differentiable at $\gamma$ and the slope of $\phi(\gamma)$ w.r.t. $\gamma$ is given by $\phi'(\gamma) = D_\gamma \mathcal{L}(\mathbf{x}(\gamma), \gamma)$.*

Now we are at the position to prove Theorem 1.

*Proof of Theorem 1.* To prove the differentiability of $\omega(\gamma)$, we apply Lemma 2 with $\mathcal{X} = \mathbb{R}^n, \mathcal{U} = \mathbb{R}$ and $\Phi = \{\mathbf{x} \in \mathcal{X} : \mathbf{0} \le \mathbf{x} \le \mathbf{1}\}$. Immediately we have that: i. $\mathcal{L}(\mathbf{x}, \cdot)$ is differentiable; ii. $\mathcal{L}(\mathbf{x}, \gamma)$ and $D_\gamma \mathcal{L}(\mathbf{x}, \gamma) = \mathbf{x}^\top \mathbf{1} - k$ are continuous on $\mathcal{X} \times \mathcal{U}$; iii. $\Phi$ is a compact subset of $\mathcal{X}$; and iv. $\forall \gamma \in \mathcal{U}$, $\mathcal{L}(\mathbf{x}, \gamma)$ has a unique minimizer $\mathbf{x}(\gamma) = \min\big(\mathbf{1}, [\mathbf{y} - \gamma\mathbf{1}]_+\big)$ over $\Phi$. The last one follows that $\nabla_\mathbf{x} \mathcal{L}(\mathbf{x}, \gamma) = \mathbf{0}$ has a unique solution and further indicates that $\mathbf{x}(\gamma) = \min\big(\mathbf{1}, [\mathbf{y} - \gamma\mathbf{1}]_+\big) = \mathrm{argmin}\, \mathcal{L}(\mathbf{x}, \gamma)$. Now applying Lemma 2 gives $\phi(\gamma) = \inf_{\mathbf{x} \in \Phi} \mathcal{L}(\mathbf{x}, \gamma) = \mathcal{L}\big(\min(\mathbf{1}, [\mathbf{y} - \gamma\mathbf{1}]_+), \gamma\big)$ is differentiable, and the slope is

$$\phi'(\gamma) = \min\big(\mathbf{1}, [\mathbf{y} - \gamma\mathbf{1}]_+\big)^\top \mathbf{1} - k = -\omega'(\gamma). \qquad (7)$$

We now show the derivative on $\omega'(\gamma)$ w.r.t. $\gamma$ is the number of element of the vector $\mathbf{v} = \mathbf{y} - \gamma\mathbf{1}$ that is strictly between 0 and 1. Let $S_0, S_1, S$ be the sets of indices $i$ that $v_i \le 0$, $v_i \ge 1$, and $0 < v_i < 1$, resp.. Now, expressing $\phi'$ in (7) using the sets $S_0, S_1, S$ gives

$$\phi'(\gamma) = \left( \sum_{i \in S_0} 0 + \sum_{i \in S_1} 1 + \sum_{i \in S} (y_i - \gamma) \right) - k = |S_1| + \left( \sum_{i \in S} y_i \right) - |S|\gamma - k, \qquad (8)$$

which gives $\phi'(\gamma) = -|S|\gamma + \text{constant}$, hence, $\omega''(\gamma) = -\phi''(\gamma) = -(-|S|)$ which gives (6).

It is clear that $\mathcal{L}(x, \gamma)$ in (2) is convex in $\mathbf{x}$ and the constraint set is closed and convex. For $(\mathbf{y}, k)$ that the feasible set for $\mathbf{x}$ is nonempty, by von Neumann Lemma [15], there exists a saddle point for Problem (2), as a result, $\phi(\gamma)$ is concave and $\omega(\gamma) = -\phi(\gamma)$ is convex.

Lastly, we show the Lipschitz constant of $\omega'(\gamma)$ is $n$: let $\mathbf{v}(\gamma) = \mathbf{y} - \gamma\mathbf{1}$, then

$$\begin{aligned}
\big| \omega'(\gamma_1) - \omega'(\gamma_2) \big| \quad &= \quad \Big\| \min\big(\mathbf{1}, \big[\mathbf{v}(\gamma_1)\big]_+\big)^\top \mathbf{1} - \min\big(\mathbf{1}, \big[\mathbf{v}(\gamma_2)\big]_+\big)^\top \mathbf{1} \Big\|_2 \\
&\le \quad \|\mathbf{1}\|_2 \cdot \Big\| \min\big(\mathbf{1}, [\mathbf{v}(\gamma_1)]_+\big) - \min\big(\mathbf{1}, [\mathbf{v}(\gamma_2)]_+\big) \Big\|_2 \\
&\overset{\text{Lemma 1}}{\le} \quad \sqrt{n}\|\mathbf{v}(\gamma_1) - \mathbf{v}(\gamma_2)\|_2 \quad = \sqrt{n}\|\gamma_1 \mathbf{1} - \gamma_2 \mathbf{1}\|_2 = n|\gamma_1 - \gamma_2|.
\end{aligned}$$

$\square$

## 2.3 Solving the projection by Newton's method

Now it is clear that we solve Problem (1) by solving the scalar minimization problem (5) using Newton's method, see Algorithm 1.

Before we discuss the case of projection onto $\Delta_k$, we first give some remarks about Algorithm 1.

**Algorithm 1:** Newton's method for solving Problem (1) onto $\Delta_k^=$

---

**1** Input: a nonzero vector $\mathbf{y}$, and initialize $\gamma_0 \in (\max(y_i) - 1, \max(y_i))$ ;

**2** **for** $t = 1, 2, ...$ **do**

**3** $\quad$ $\mathbf{v} = \mathbf{y} - \gamma_{t-1}\mathbf{1}$;

**4** $\quad$ $\gamma_t = \gamma_{t-1} - \dfrac{\omega'(\gamma_{t-1})}{\omega''(\gamma_{t-1})}$, for $\omega'(\gamma_{t-1}) = k - \min\left(\mathbf{1}, [\mathbf{v}]_+\right)^\top \mathbf{1}$, $\omega''(\gamma_{t-1}) = \displaystyle\sum_{i=1}^{n} I_{0 < v_i < 1}$;

**5** **end**

**6** **return** $\mathbf{x}^* = \min(\mathbf{1}, [\mathbf{v}]_+)$

---

**Quantify the performance of the algorithm: feasibility gap** As the goal of solving Problem (1) is to find a vector inside the $k$-crapped simplex $\Delta_k^=$, hence we can define the feasibility gap as

$$\text{dist}(\mathbf{x}, \Delta_k^=)^2 := \text{dist}\left(\mathbf{x}, \left\{\mathbf{u} \in \mathbb{R}^n \mid \mathbf{u}^\top \mathbf{1} = k\right\}\right)^2 + \text{dist}\left(\mathbf{x}, \left\{\mathbf{u} \in \mathbb{R}^n \mid \mathbf{0} \leq \mathbf{u} \leq \mathbf{1}\right\}\right)^2.$$

By the definition of the algorithm, $\mathbf{x}^*$ is always inside the interval $[\mathbf{0}, \mathbf{1}]$, so we can simplify the feasibility gap as $\text{dist}\left(\mathbf{x}, \left\{\mathbf{u} \in \mathbb{R}^n \mid \mathbf{u}^\top \mathbf{1} = k\right\}\right) = |\mathbf{x}^\top \mathbf{1} - k|$. For the toy example in Fig. 1, we can see that such a gap is monotonically decreasing for the proposed algorithm.

**Range for $\gamma^*$ and $\omega''(\gamma^*)$ for general problem** In general, the minimizer $\gamma^*$ can take any value in $[-\infty, +\infty]$ (with $\infty$ included) for any $(\mathbf{y}, k)$. Also, by definition, $\omega''(\gamma^*)$ can take any value in $[0, n]$. Hence, cases like $\omega''(\gamma^*) = 0$ and/or $\gamma^* \in \{\pm\infty\}$ are possible. For example, suppose $k = n$, then for any bounded $\mathbf{y} \in \mathbb{R}^n$, one can find that a root of $\omega'(\gamma) = \omega''(\gamma) = 0$ is $\gamma = -\infty$. Furthermore, if the parameter $k$ is too large (or too small) such that $l_1$ and $l_2$ in Fig. 1 do not intersect each other, it is possible for the algorithm to produce $\gamma^*$ with $\omega''(\gamma^*) = 0$ and/or $\gamma* \in \{\pm\infty\}$.

**On $k$, and the feasible range of $\gamma$** In practice, $k$ is a user-defined value that is related to the sparsity of the desired solution vector, so $k$ is usually (much) smaller than $n$, therefore we focus on the case $k \ll n$ (in § 4, we will illustrate the experimental result on solving bioinformatics problems on real-world datasets, where we used $k \ll n$). In the following, we discuss $\gamma$, assuming $0 < k \ll n$ is selected such that the Problem (1) is feasible to solve. Now we use the toy example in Fig. 1 as an illustration. As Problem (1) is feasible, we have $l_1$ and $l_2$ in Fig. 1 intersect at a $\gamma^* \notin \{\pm\infty\}$ (in the toy example, $\gamma^*$ is unique). Now for the update of $\gamma$ in Algorithm 1 to work, we need $\omega''(\gamma) \neq 0$. Notice that $\omega''(\gamma) = 0 \iff \{v_j \leq 0 \text{ or } v_j \geq 1\} \,\forall j$, i.e., the "bad region" such that $\omega''(\gamma) = 0$ is the intersection of all $\mathcal{I}_j := \{\gamma \leq y_j - 1 \text{ or } y_j \leq \gamma\}$. Therefore, the feasible range for $\gamma$ such that $\omega''(\gamma) \neq 0$ is the complement of such bad region, i.e,

$$\gamma \in \mathbb{R} \backslash \bigcap_j \mathcal{I}_j \subset \left]\min(y_i) - 1, \max(y_i)\right[, \tag{9}$$

where $]a, b[$ denotes the open interval $a < x < b$, and $\min(y_i), \max(y_i)$ are the smallest and largest value in $\mathbf{y}$, resp.. We can make use of this to design an initialization of $\gamma_0$. For example, we can simply set $\gamma_0 \in [\min(y_i) - 1 + \delta, \min(y_i) - \delta]$ for a small constant $\delta > 0$, which has the cost of $\mathcal{O}(1)$. Or, we can search $\gamma_0$ within the interval (9), with a worst-case complexity of $\mathcal{O}(n \log n)$, by sorting all the $n$ elements of $y_i$ and trying all the $n$ feasible intervals. Refer to the toy example in Fig. 1, we see that once $\gamma_0$ is initialized within the feasible range, the algorithm will produce a sequence $\{\gamma_t\}_{t \in \mathbb{N}}$ that converges to the minimizer $\gamma^*$. The convergence is guaranteed based on the convergence theory of Newton's method. Furthermore, if $\gamma^* \neq \infty$, the sequence $\{\gamma_t\}_{t \in \mathbb{N}}$ will not diverges to infinity since all $\omega''(\gamma) \neq 0$, based on the following corollary.

**Corollary 2.** *If $\gamma^* \neq \infty$ is the minimizer of Problem (5), then $\omega''(\gamma^*) \neq 0$.*

*Proof.* As $\omega'(\gamma^*) = 0$, then $\gamma^* \stackrel{(7)}{=} \frac{k - |S_1| + \sum_{i \in S} y_i}{|S|} \neq 0$ and hence $|S| \neq 0$ and $\omega''(\gamma^*) \neq 0$. $\quad\square$

**Computational cost** The cost of the Algorithm 1 mainly comes from line 4, which has a per-iteration cost of $\mathcal{O}(n)$. Assuming it takes the algorithm $T$ iterations to converge, then the total cost inside the loop is $T\mathcal{O}(n)$. Theoretically speaking, it can take Newton's method many iterations to converge, but empirically we observed that the algorithm usually converges in a small number of iterations, so $T$ is small, see Table 2 in § 3. In comparison, it costs $\mathcal{O}(n^2)$ for the method proposed in [17] for solving the root of the piecewise-linear equation $\omega'(\gamma) = 0$.

**Comparisons with existing methods**   To the best of our knowledge, all existing works solve the projection problem (or a similar type of projection problem) by finding the root of $\omega'(\gamma) = 0$ [4, 5, 17][3], in which they solve such a piecewise-linear equation by sorting-based techniques. As these methods do solve $\omega'(\gamma) = 0$ exactly, so when the projection problem is feasible, they will eventually produce an exact result. In contrast, the proposed method is an iterative numerical algorithm, so it is possible that, due to numerical issues, the proposed method does not produce an exact solution but only an approximate solution. However, we argue that, for application with an input vector that has a huge dimension, the proposed method is good enough.

If the equation $\omega'(\gamma) = 0$ turns out to have no root for the input pair $(\mathbf{y}, k)$, in this case, all the sorting-based methods will fail, but the proposed algorithm is still able to produce an approximate solution, which can be useful in practice.

Lastly, as the proposed method has a lower computational cost, the method is very suitable for the projection of high dimensional vectors, see § 3. The proposed method also finds application on large-scale bioinformatics problems, see § 4.

**The case of projection onto $\Delta_k$**   If we want to project onto $\Delta_k$, the same algorithm can be used with the following twists: i. The feasibility gap becomes

$$\text{dist}\Big(\mathbf{x}, \big\{\mathbf{u} \in \mathbb{R}^n \mid \mathbf{u}^\top \mathbf{1} \le k\big\}\Big) = \begin{cases} 0 & \text{if } \mathbf{x}^\top \mathbf{1} \le k \\ \mathbf{x}^\top \mathbf{1} - k & \text{else} \end{cases}, \text{ and}$$

ii. $\gamma$ has to be nonnegative, which can be done simply as $\gamma = [\gamma]_+$. Note that now it is possible for $\omega''(\gamma) = 0$, since the safe range (9) may become empty after $\gamma = [\gamma]_+$, for some input vector $\mathbf{y}$.

## 3   Experiment on comparing the projection algorithms

Now we show the experimental results to demonstrate the efficiency of the proposed projection algorithm. The algorithm is implemented in MATLAB[4], and is compared with the MATLAB command quadprog (a MATLAB general-purpose solver for the quadratic program), the C++ implementation of [17], and a state-of-the-art commercial solver Gurobi [7] that uses the interior-point method [9] to solve the projection problem. All experiments were conducted on a PC with an Intel Xeon CPU (3.7GHz) and 32GB memory.

**The result of Table 1: fast convergence of the proposed method across data size**   We compared the methods on the simulation data used in [17]. We generate vector $\mathbf{y} \in \mathbb{R}^n$ with $y_i$ uniformly sampled from $[-0.5, 0.5]$. We pick $k$ as an integer between 1 and $n$ by random, and we pick the dimension $n \in \{50, 10^2, 10^3, 10^4, 10^5, 10^6, 10^7, 10^8\}$. We repeat the experiments 100 times, and the comparison results are shown in Table 1. We see that as $n$ increases, the efficiency of the proposed algorithm becomes more and more significant. For large $n$, many methods take more than 60 seconds (denoted as '-' in Table 1). The results confirmed that the proposed projection method is the most efficient one among these methods. In the following experiments, we only compare the proposed algorithm and the commercial solver Gurobi, as they are the two most efficient methods.

**The result of Table 2: it only takes a small number of iterations for the algorithm to converge**
We currently do not have a solid theory on characterizing the exact convergent rate for Algorithm 1, hence we performed experiments to showcase the relationship between $T$ (the number of iterations for Algorithm 1 to converge) and $n$ (the number of dimension of the input vector $\mathbf{y}$). From Table 2, we see an expected result that runtime and $T$ increase with $n$, and the table empirically supports the claim that it only takes a small number of iterations for the proposed method to converge. For example, for $n = 10^6$, the empirical complexity of the proposed approach is about $10\mathcal{O}(n)$, which is much lower than $\mathcal{O}(n \log n)$ and $\mathcal{O}(n^2)$ with such a large $n$.

**The result of Fig. 2: fast convergence of the proposed method across range**   We compare the performance of the methods on solving the projection problem w.r.t. different range of the elements of $\mathbf{y}$. Here, we generate $\mathbf{y} \in \mathbb{R}^n$ uniformly from $[-\alpha, +\alpha]$ with $n = 10^5$ and $\alpha = \{1, 10, 100, 1000\}$. As shown in Fig. 2, the proposed algorithm converges much faster than the commercial solver. The figure also shows that the performance of the proposed method is not sensitive to $\alpha$.

---

[3]Note that [4, 5] do not tackle Problem (1) as they only solve the projection onto the unit simplex.
[4]The code is in the appendix.

Table 1: Runtime in seconds (in mean±std) of different methods on various sizes $n$.

| Method | $5 \times 10^1$ | $10^2$ | $10^3$ | $10^4$ |
|---|---|---|---|---|
| quadprog | $0.0128 \pm 0.01058$ | $0.0034 \pm 0.0095$ | $0.1419 \pm 0.0183$ | - |
| C++ | $\mathbf{0.00003 \pm 0.00007}$ | $\mathbf{0.00004 \pm 0.00004}$ | $0.0011 \pm 0.0014$ | $0.10 \pm 0.129$ |
| Gurobi | $0.002 \pm 0.0048$ | $0.0022 \pm 0.0021$ | $0.0041 \pm 0.0022$ | $0.02 \pm 0.0032$ |
| Proposed | $0.00013 \pm 0.00014$ | $0.00016 \pm 0.00065$ | $\mathbf{0.00039 \pm 0.00038}$ | $\mathbf{0.0039 \pm 0.0037}$ |

| Method | $10^5$ | $10^6$ | $10^7$ | $10^8$ |
|---|---|---|---|---|
| quadprog | - | - | - | - |
| C++ | $10.3985 \pm 13.703$ | - | - | - |
| Gurobi | $0.2413 \pm 0.00117$ | $2.8229 \pm 0.0955$ | $32.26 \pm 0.4953$ | - |
| Proposed | $\mathbf{0.0406 \pm 0.0377}$ | $\mathbf{0.4723 \pm 0.3629}$ | $\mathbf{4.2067 \pm 2.5795}$ | $\mathbf{14.925 \pm 2.9893}$ |

Table 2: Runtime and # iterations (in mean) on varying $n, k$ (over 100 experiments).

| Varying $n$ $(k = 10^2)$ | Time | # iter. $T$ | Varying $k$ $(n = 10^6)$ | Time | # iter. $T$ |
|---|---|---|---|---|---|
| $n = 10^4$ | $0.0011 \pm 0.00013$ | $6.2 \pm 0.4$ | $k = 10^1$ | $0.15 \pm 0.0096$ | $11.4 \pm 0.49$ |
| $n = 10^5$ | $0.0108 \pm 0.00081$ | $8.0 \pm 0.2$ | $k = 10^2$ | $0.13 \pm 0.0054$ | $10 \pm 0$ |
| $n = 10^6$ | $0.13 \pm 0.0054$ | $10 \pm 0$ | $k = 10^3$ | $0.12 \pm 0.0084$ | $8.5 \pm 0.5$ |
| $n = 10^7$ | $2.1 \pm 0.03$ | $12 \pm 0$ | $k = 10^4$ | $0.11 \pm 0.0039$ | $7 \pm 0$ |
| $n = 10^8$ | $21.95 \pm 0.046$ | $13 \pm 0$ | $k = 10^5$ | $0.10 \pm 0.0065$ | $5.1 \pm 0.2$ |

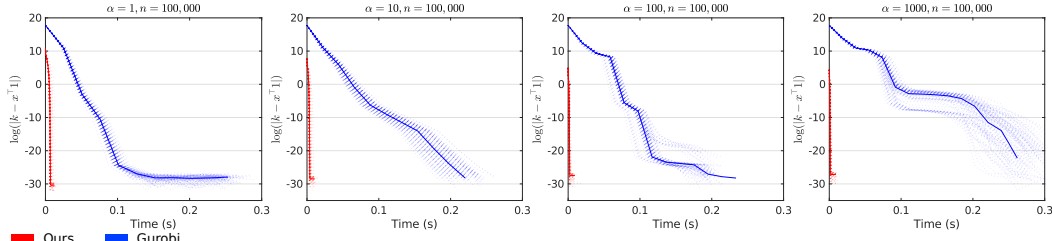

Figure 2: Comparison between the proposed projection method and the Gurobi projection for $\alpha \in \{1, 10, 100, 1000\}$. The thick curves are the median of the results of over 100 experiments on 100 datasets. The figure shows the superior performance of the proposed method in terms of runtime.

## 4 Application of the projection in sparse regression

An application of the proposed projection method is solving the cardinality-constrained $\ell_2^2$-regularized regression [8]:

$$\min_{\|\mathbf{w}\|_0 \leq k} \left\{ F(\mathbf{w}) := \sum_{i=1}^{n} (y_i - \mathbf{w}^\top \mathbf{x}_i)^2 + \frac{1}{2} \rho \|\mathbf{w}\|_2^2 \right\}, \qquad (P^*)$$

where $\left\{ (\mathbf{x}_i, y_i) \in \mathbb{R}^n \times \mathbb{R} \right\}_{i=1}^{m}$ is a collection of $m$ samples of data point with label. The sparsity-inducing constraint $\|\mathbf{w}\|_0 \leq k$ enforces $\mathbf{w} \in \mathbb{R}^n$ to be at most $k$-sparse (with $k$ non-zero elements). A recent work [10] studied the Boolean relaxation of $P^*$, which gives the following problem

$$\min_{\mathbf{u} \in \Delta_k} \left\{ G(\mathbf{u}) := \mathbf{y}^\top \left( \frac{1}{\rho} \mathbf{X} \mathbf{D}(\mathbf{u}) \mathbf{X}^\top + \mathbf{I} \right)^{-1} \mathbf{y} \right\}, \qquad (P_{\text{BR}})$$

where $\mathbf{u} \in \Delta_k$ means $\mathbf{u}$ has to be inside the $k$-capped simplex. Both the original work of [10] and several follow-up papers [1, 2] showed that the solution of the Boolean relaxation $P_{\text{BR}}$ empirically outperforms the solution of other sparse estimation methods on recovering the sparse features, such as Lasso [14] and elastic net [18], especially when the sample size is small and the feature dimension is huge. These empirical results inspired us to develop an efficient and scalable algorithm for solving $P_{\text{BR}}$, which in turn motivates the study of solving Problem (1).

Now we show extensive experimental results to demonstrate the efficiency of the proposed method (Algorithm 1) on accelerating the Projected Quasi-Newton (PQN) method [11] on solving $P_{\text{BR}}$ on big

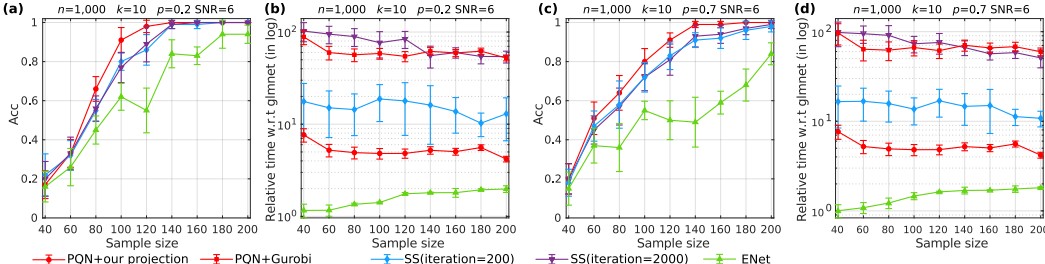

Figure 3: Comparison between different methods. The results are the average over 100 datasets, and all the error bars are in mean±std. From left to right, the sub-figures are: (a) The Acc values between different methods when $n = 10^3$, $k = 10$, $p = 0.2$, and SNR= 6; (b) The computation time of the algorithms in (a); (c) The Acc values between different methods when $p$ changed to 0.7; and (d) The computation time of the algorithms in (c).

data. To be specific, we use both a simulation dataset and a real-world dataset with various sizes. We compare the method with a state-of-the-art method, which uses a subgradient method [2] (denoted as SS) implemented in Julia to solve a min-max problem that is equivalent to $P_{\text{BR}}$. In addition, we compare with the classical elastic net (ENet) [13] as a reference. For a fair comparison between methods that are implemented in different programming languages, we report the computational time for each algorithm relative to the time needed to compute a Lasso estimator with Glmnet [6, 13] on the same programming language, and on the same dataset. Glmnet is a package that fits the data to a generalized linear model, based on a penalized maximum likelihood estimation method. Glmnet has been implemented in various programming languages, such as Julia, R, and MATLAB.

**Sparse regression on simulation data**  First, we conduct experiments on simulation data generated as follows: let $[m] = \{1, 2, \ldots, m\}$, we draw $\mathbf{x}_i \sim \mathcal{N}(\mathbf{0}_n, \mathbf{\Sigma})$, $i \in [m]$ independently from a $n$-dimensional normal distribution with zero mean and co-variance matrix $\mathbf{\Sigma}$. We randomly sample a weight vector $\mathbf{w}^{\text{true}} \in \{-1, 0, +1\}$ with exactly $k_{\text{true}}$ non-zero coefficients. Then we generate noise vector $\epsilon \in \mathbb{R}^m$, where $\epsilon_i$, $i \in [m]$ are drawn independently from a normal distribution scaled according to a chosen signal-to-noise ratio (SNR) defined as $\sqrt{\text{SNR}} = \|\mathbf{X}\mathbf{w}^{\text{true}}\| / \|\epsilon\|$. Finally, we form $\mathbf{Y} = \mathbf{X}\mathbf{w}^{\text{true}} + \epsilon$, where $\mathbf{X} = [\mathbf{x}_1, \mathbf{x}_2, \ldots, \mathbf{x}_n]^\top$. We evaluate the performance on simulation data using accuracy defined as $\text{Acc}(\mathbf{w}) := \left|\{i : w_i \neq 0, w_i^{\text{true}} \neq 0\}\right| / \left|\{i : w_i^{\text{true}} \neq 0\}\right|$.

For the first simulation data, we generate 100 datasets by setting $\mathbf{\Sigma}_{ij} = p^{|i-j|}$ to a Toeplitz covariance matrix with $p = 0.2$, $n = 1,000$, $k_{\text{true}} = 20$, and SNR= 6. As shown in Fig. 3 (a), PQN + the proposed projection and PQN + Gurobi have exactly the same performance in in terms of Acc value, which is expected since the only difference between these methods is the projection algorithm. In addition, the two methods converge quickly to Acc= 1, outperforming all the other methods. Fig. 3 (b) shows that PQN + the proposed projection is the most efficient algorithm for solving $P_{\text{BR}}$ among all the tested methods.

The second simulation data is generated similarly to the first one. The only difference here is that $p = 0.7$, indicting the higher correlation between the features, meaning that now it is harder to recover the true support of $\mathbf{w}^{\text{true}}$. As shown in Fig. 3 (c), PQN + the proposed projection and PQN + Gurobi again have the same performance, and they still outperform all the other methods. Again, PQN + the proposed projection is the most efficient algorithm for solving the problem here.

For the results on comparing the convergence between PQN and SS, as well as the details of all the parameter settings for all the methods, see the appendix.

**Sparse regression on GWAS data**  We conducted experiments to evaluate the efficiency of the proposed projection algorithm on a Genome-Wide Association Study (GWAS) dataset. The data was obtained from the dbGaP web site, under the dbGaP Study Accession phs000095.v3.p1[5]. This study is part of the Gene Environment Association Studies initiative with the goal to identify novel genetic factors that contribute to dental caries through large-scale genome-wide association studies of well-characterized families and individuals at multiple sites in the United States. The data we used in this experiment contains 227 Caucasians with 11,626,696 Single-nucleotide polymorphisms

---

[5] https://www.ncbi.nlm.nih.gov/projects/gap/cgi-bin/study.cgi?study_id=phs000095.v3.p1

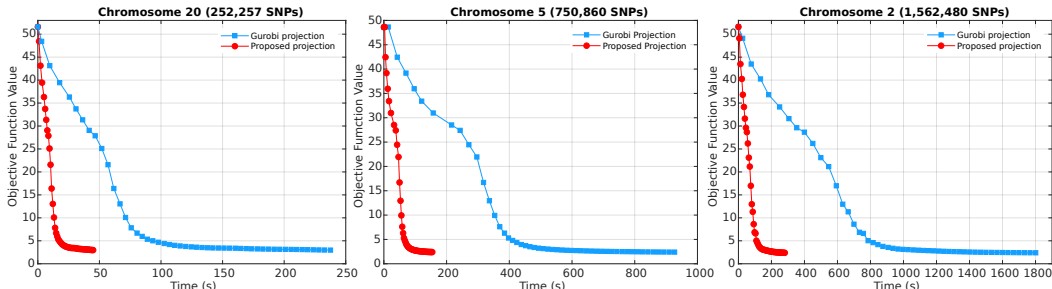

Figure 4: Convergence comparison between PQN with the proposed projection algorithm and PQN with Gurobi projection. From left to right, the sub-figures are: (a) Comparison on chromosome 20; (b)Comparison on chromosome 5; and (c) Comparison on chromosome 2.

(SNPs) for all chromosomes. Here, the goal is to identify the important SNPs by performing a sparse regression. We solve the Boolean-relaxed sparse regression problem $P_{\text{BR}}$ on chromosome 20, 5, and 2, which have 252,257 SNPs, 750,860 SNPs, and 1,562,480 SNPs, resp..

Fig. 4 shows the comparison result between PQN + the projection and PQN + Gurobi. We find that PQN + the proposed projection outperforms in all three problems (with different numbers of SNPs) in terms of computational time. In addition, PQN + the proposed projection and PQN + Gurobi converge to the same objective function value, indicating the correctness of the proposed projection algorithm.

**Results on comparing different methods on the simulated GWAS data**    To further demonstrate the practical impact of solving $P_{\text{BR}}$ using the proposed algorithm, here we further compare different methods using the simulated GWAS data described in [16].

We compared $P_{\text{BR}}$ with Lasso [14] and Elastic Net (ENet). We used the out-of-sample Mean Square Error (MSE) to select the parameter $k$ for $P_{\text{BR}}$. For Lasso and ENet, we used cross-validation with the minimum MES+1SE (Standard Error) to tune the hyper-parameters, which was suggested by [16]. There are three different simulated GWAS datasets as stated in [16], in which they have different levels of correlation (LD: High, Mixed, and Low) between SNPs. All the datasets have 50,000 SNPs and 25 significant SNPs. In Table 3, we showed the comparison when the sample size is 1000 and 100. Here we report the median and the standard deviation (in parentheses) over 100 replicates. We can see that $P_{\text{BR}}$ can find more correct SNPs with low false positive rate, especially when the sample size is only 100. For the comparisons when the sample size is 150 and 50, see the appendix.

Table 3: SNP=50,000, significant SNPs = 25.

|  |  | Sample size = 1000 | | | Sample size = 100 | | |
|---|---|---|---|---|---|---|---|
|  |  | Lasso | ENet | $P_{\text{BR}}$ | Lasso | ENet | $P_{\text{BR}}$ |
| High LD | Correct | 3(0.91) | 25(0) | 25(0) | 2(1.01) | 20(3.47) | 25(1.52) |
|  | False positive | 0(0) | 0(0.1) | 0(0) | 1(0.93) | 4(1.41) | 1(1.03) |
| Mixed LD | Correct | 3(0.75) | 18(1.47) | 25(0) | 2(1.35) | 16(1.47) | 24(2.03) |
|  | False positive | 0(0) | 0(1.19) | 0(0) | 1(0.68) | 2(0.35) | 2(1.24) |
| Low LD | Correct | 17(2.12) | 25(0.51) | 25(0) | 5(2.12) | 17(3.51) | 23(3.07) |
|  | False positive | 0(0.39) | 0(2.06) | 0(0) | 2(0.59) | 5(2.06) | 3(1.47) |

## 5    Conclusion

We propose to use Newton's method to solve the problem of projecting a vector onto the $k$-capped simplex. On the theory side, we transform the problem to a scalar minimization problem and provide a theorem to characterize the derivatives of the cost function of such a problem. On the practical side, we show that the proposed method outperforms existing methods on solving the projection problem, and show that it can be used to accelerate the method on solving sparse regressions in bioinformatics on a huge dataset.

## 6    Acknowledgments

The presented materials are based upon the research supported by the Indiana University's Precision Health Initiative. Andersen Ang acknowledges the support by a grant from NSERC (Natural Sciences and Engineering Research Council) of Canada.

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
