# A  Appendix

## A.1  About the reference method ENet on the experiment in Section 4

ENet is the following regularized least squares model [14]:

$$\min_{\mathbf{w}\in\mathbb{R}^n}\left\{\sum_{i=1}^{n}(y_i-\mathbf{w}^\top\mathbf{x}_i)^2+\lambda_1\|\mathbf{w}\|_1+\lambda_2\|\mathbf{w}\|_2^2\right\},$$

where $\lambda_1$ and $\lambda_2$ are regularization parameters. The authors solve this problem by an algorithm called LARS-EN [14], which we will abuse the notation and refer it also as ENet below.

## A.2  Optimization details in Section 4

We now give more details on the experiments discussed in Section 4. First, we briefly recall about the methods that we compare.

### A.2.1  The three methods: PQN and SS

First, both the PQN + the proposed projection and PQN + Gurobi solve Problem ($P_{\text{BR}}$), where they differ in the projection method. For the details of PQN, we refer to [11] (Algorithm 1) and the book chapter [12]. Next, the SS method solves a re-formulation that is equivalent to ($P_{\text{BR}}$) [2]. Specifically, the SS method, which is a dual sub-gradient type method, aims to solve the following quadratic min-max problem

$$\min_{\mathbf{u}\in\Delta_k}\max_{\mathbf{v}\in\mathbb{R}^m}\left\{-\frac{1}{2}\mathbf{v}^\top\left(\frac{1}{\rho}\mathbf{X}\mathbf{D}(\mathbf{u})\mathbf{X}^\top+\mathbf{I}\right)\mathbf{v}-\mathbf{v}^\top\mathbf{y}\right\},\qquad(P_{\text{BR}}^*)$$

where $\mathbf{v}$ is a dual variable to $\mathbf{w}$. It is important to note that, the problems ($P_{\text{BR}}$) and ($P_{\text{BR}}^*$) are shown to be equivalent [2, 10].

### A.2.2  The experimental settings

**On simulation datasets**  For comparison on simulation datasets, PQN + the proposed projection and PQN + Gurobi share the same initialization, which is $\mathbf{u}_0=\frac{k}{n}\mathbf{1}$. It is trivial to verify that $\mathbf{u}_0\in\Delta_k$. Following the implementations, the SS method and the reference model ENet are both randomly initialized [2, 14]. We let both PQN + the proposed projection and PQN + Gurobi run 50 iterations, and we let SS method run 200 (as suggested in [2]) and 2,000 iterations.

There are two hyper-parameters $\rho$ and $k$ for $P^*$ and $P_{\text{BR}}$. We set $\rho=\frac{1}{\sqrt{m}}$ as recommend in [2] for PQN + the proposed projection, PQN+Gurobi, and the SS method. We assume that we know $k=k_{\text{true}}$ as the ground-truth. Specifically, in Fig. 3, $k=20$ is used for all the experiments. For ENet, which do not have the parameter $k$, we screen through the values of $\lambda_1$ and $\lambda_2$ such that the output vector $\mathbf{w}$ has exactly $k=20$ non-zero elements.

**On real-world datasets**  For the comparison on GWAS datasets, we initialize PQN + the proposed projection and PQN + Gurobi by setting $\mathbf{u}_0=\frac{k}{n}\mathbf{1}$. We let both PQN + the proposed projection and PQN + Gurobi run 50 iterations. Furthermore, we set $\rho=\frac{1}{\sqrt{m}}$ and $k=100$ for both methods. We emphasize that these two methods share exactly the same setting in the experiments, while their difference is only on the way they solve the projection problem.

## A.3  Convergence Comparison on Solving $P_{\text{BR}}$

In this section, we compare the convergence between PQN + the proposed projection, PQN + Gurobi, and the SS method. PQN + the proposed projection and PQN + Gurobi solve Problem ($P_{\text{BR}}$), and the SS method solves an equivalent re-formulation $P_{\text{BR}}^*$. As shown in Fig. 3 (a) and (c) in the main text, the PQN methods have a better performances on discovering the correct features than the SS method. This can be explained by Fig 5 (a): clearly, although PQN + the proposed projection, PQN + Gurobi, and the SS method are solving the same problem, they have very difference convergence performance. PQN + the proposed projection and PQN + Gurobi converge to the same objective function value, which is much lower than that of the SS method. We emphasize that, this has been observed in all of our experiments.

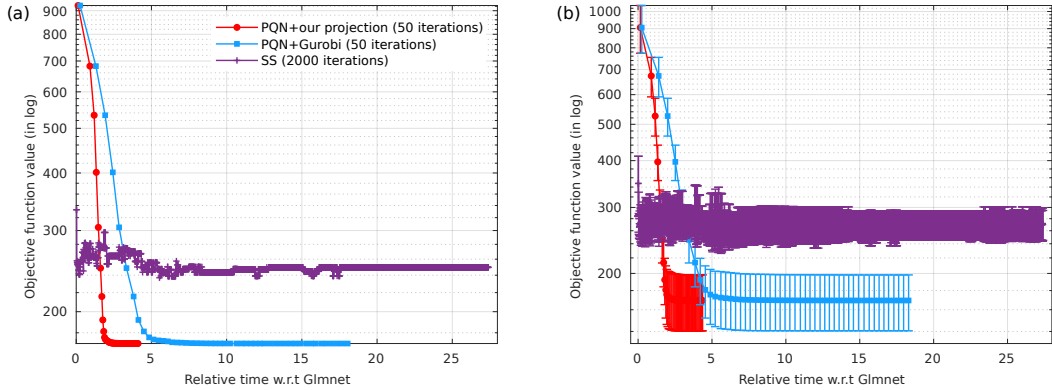

Figure 5: Convergence comparison (plotted in error bar) between PQN + our projection, PQN + Gurobi, and the SS method. (a) Convergence comparison on a one dataset. (b) Converge comparison on 10 different datasets.

Then, we run these three methods on 10 different simulation datasets generated by the procedure described in § 4 ($n = 10^3$, $k = 10$, $p = 0.2$, $m = 100$, and SNR= 6), see Fig.5 (b) for the convergence performance. We have the consistent observation that, PQN + the proposed projection and PQN + Gurobi converge to lower objective function values than the SS method. The convergence performances (as shown in Fig.5) give the reason why, PQN + the proposed projection and PQN + Gurobi have a better performance in accuracy on finding the features (as shown in Fig.3) than the SS method.

## A.4 Further results on comparing different methods on the simulated GWAS data

The following table gives further results corresponding to Table 3 in the main text.

Table 4: SNP=50,000, significant SNPs = 25.

| | | Sample size = 150 | | | Sample size = 50 | | |
|---|---|---|---|---|---|---|---|
| | | Lasso | ENet | $P_{BR}$ | Lasso | ENet | $P_{BR}$ |
| High LD | Correct | 2(1.91) | 24(1.37) | 25(0.46) | 2(1.21) | 17(4.47) | 19(3.52) |
| | False positive | 1(0.92) | 10(1.92) | 1(0.47) | 4(1.93) | 14(3.41) | 5(2.03) |
| Mixed LD | Correct | 3(1.75) | 19(1.87) | 24(0.87) | 2(1.35) | 12(2.47) | 16(2.23) |
| | False positive | 1(0.63) | 16(1.09) | 1(0.24) | 2(1.68) | 8(2.35) | 6(2.24) |
| Low LD | Correct | 7(1.23) | 20(0.81) | 25(1.26) | 1(0.32) | 12(3.61) | 15(3.27) |
| | False positive | 3(0.72) | 18(1.26) | 1(0.27) | 0(0.19) | 31(5.06) | 10(3.47) |

## A.5 The MATLAB codes for the projection

The codes are available at https://anonymous.4open.science/r/Projection_Capped_Simplex-BB7C.