# OpenReview forum: "Fast Projection onto the Capped Simplex with Applications to Sparse Regression in Bioinformatics"
_NeurIPS.cc/2021/Conference — NeurIPS 2021 Poster_

### Official Review · Reviewer_Rv7X · 2021-07-16

**Rating:** 6
**Confidence:** 3

**Summary:**

This paper addresses the problem of accelerating the projection of a vector onto the k-capped simplex. From the theoretical side, the authors showed that the given problem can be casted as a scalar minimization problem, which is tackled by a Newton’s method, unlike existing methods based on sorting algorithm.
In computational experiments, the authors evaluated the proposed method in a sparse regression problem. The baselines are Matlab’s quadprog, reference[15], and Gurobi, a commercial interior point solver.
The proposed method worked faster especially when the response vector y is high-dimensional. It is also shown to have a higher support recovery nature than that of Elastic net. In experiments using real dataset consisting up to 1.5 million SNPs, the proposed method worked much faster than Gurobi.

**Limitations And Societal Impact:**

The authors claimed that the computational complexity of the proposed method is T O(n), where T is the number of Newton iterations, and only a few iterations were required empirically.
However, this issue should be surveyed more, since linear time computational complexity is supposed to be one of the main contributions of this paper. An analysis about the worst case or average case running times should be studied.

Figure 1 is not illustrative enough due to the lack of explanation both in the caption and the main text.

#Typos
In abstract, CUP -> CPU

In section 2.1 rootof -> “root of”

In Theorem 1, convex -> concave ? since the second derivative is nonnegative.

In Theorem 1, “a indicator” -> “an indicator”

**Main Review:**

The paper is easy to read, and the theoretical contribution is fair, and the computational experiments are carried out well.

**Time Spent Reviewing:**

4

---

> ### Author Response · Authors · 2021-08-10
> **Response to Reviewer Rv7X**
>
> We thank the reviewer for the comments and the positive feedback. We also thank the reviewer for pointing out typos in the paper.
>
> For the relationship between T and n, we did an extra experiment to show their relationship empirically. Currently, we do not have a solid theory on characterizing the exact convergent rate and this will be our future work. The results are shown in Table 1: for different n, the proposed projection algorithm only takes a few iterations to converge. We did another experiment to show the impact of k as suggested by reviewer CRDH (Table 2). Combining these two tables, we find that the run time is influenced by both n and k. The projection algorithm tends to take more time for large n and small k. We will include these two tables in the next version of the paper.
>
> Table 1. Run time and #. Iterations on varying n over 100 experiments.
>
>  |        |              | Time in sec (mean (std)) | # Itr. (mean(std))
>   |---------- |----------|----------|---------|
> | n=10^4 | k=10^2 | 0.0011(0.00013)            | 6.2(0.4)
>  |n=10^5 | k=10^2 | 0.0108(0.00081)            | 8.0(0.2)
> | n=10^6 | k=10^2 | 0.13(0.0054)                   | 10(0)
> | n=10^7 | k=10^2 | 2.1(0.03)                          | 12(0)
> | n=10^8 | k=10^2 | 21.95(0.046)                   |13(0)
>
>
> Table 2. Run time and #. Iterations on varying k over 100 experiments.
>
>  |               |             | Time in sec (mean (std))   | # Itr. (mean(std)) |
>   |---------- |----------|----------|---------|
>  |n=10^6 | k=10^2 | 0.13(0.0054)                   | 10(0)    |
>  |n=10^6 | k=10^3 | 0.12(0.0084)                   | 8.5(0.5)    |
>  |n=10^6 | k=10^4 | 0.11(0.0039)                   | 7(0)          |
>  |n=10^6 | k=10^5 | 0.10(0.0065)                   | 5.1(0.2) |

---

### Official Review · Reviewer_CRDH · 2021-07-16

**Rating:** 7
**Confidence:** 3

**Summary:**

This paper proposes an algorithm based on Newton’s method for projecting a vector with bounded elements onto a hyper-cube cut by a hyperplane (also known as the k-capped simplex). The time complexity of the algorithm is roughly linear (in terms of the vector dimension) and theoretical insights for partial justification of the method are provided. Empirical evaluation demonstrates that the algorithm (1) finds a solution with high precision on large-scale datasets and (2) considerably outperforms state-of-the-art sorting-based methods in terms of run time. Finally, the proposed algorithm is also applied to solving a sparse regression problem from the bioinformatics domain on which it is able to accelerate the Projected Quasi-Newton (PQN) method from 3 to 6 times compared to its alternatives.

**Limitations And Societal Impact:**

One major concern is that it was not thoroughly examined if (and possibly how) $T$ is dependent on $n$, such investigation would be beneficial for the overall study.

In the experiments on simulated data, the Acc metric is defined so as to measure only the ‘overall’ misestimation of the $\omega_i$ values. However, $\omega_i$ can either be -1, 0, or 1, and the zero values in the entire $\mathbf{w}$ vector are predominant I presume. If this is the case, it would be insightful to measure metrics such precision or recall w.r.t. to each of the possible values in $\mathbf{w}$. Micro/macro or weighted variants of these metrics might also be applicable.

The input parameter $k$ seems to be fixed in the simulated data experiments. I am wondering if such a parameter has an impact on run time of the proposed algorithm, which should have been analyzed under different feature dimensions. If such analysis was indeed conducted, I would encourage the authors to include and discuss the findings in the paper.

**Main Review:**

Originality: The formulation of the optimization problem is well known; however, to the best of my knowledge, the idea of converting this problem into a scalar minimization problem and solving it by Newton’s iteration appears to be novel. Also, there are some theoretical insights supporting the proposed algorithm which I believe have not been presented in the literature. The related work appears to be comprehensive enough.

Quality: The design and justifications for the proposed algorithm are technically sound. The observations made regarding the high precision and efficiency of the algorithm are empirically well supported. Overall, the work seems to be mature in terms of quality.

Clarity: The paper is well written, organized, and technically detailed to a satisfactory extent. The notation is clear and consistent throughout the paper.

Significance: The problem addressed in this paper is of considerable importance as high-impact bioinformatics applications such as the sparse regression problem of identifying single-nucleotide polymorphisms for chromosomes from the Genome-Wide Association Study (GWAS) dataset. Another significant aspect of this paper is the proposed algorithm’s superior efficiency over state-of-the-art methods, while being even 6-8 times more efficient than commercial software on highly dimensional data with million variables or more.


**Time Spent Reviewing:**

5

---

> ### Author Response · Authors · 2021-08-10
> **Response to Reviewer CRDH**
>
> First, we thank the reviewer for the comments and the positive feedback. We agree that we did not express enough how T  depends on n. So we provide a table below to showcase the relationship between T and n.
>
> From Table 1, we see that the run time and the # Iteration increases with n, which we currently do not have a solid theory on characterizing the exact convergent rate and this will be our future work. The table also empirically supports our claim in the paper that it only takes a few iterations for the proposed method to converge. For example, for n = 10^6, the empirical complexity of the proposed approach is about 10 $O(n)$, which is much lower than $O(n log n)$ and $O(n^2)$ with such as large n.
>
> Table 1. Run time and #. Iterations on varying n over 100 experiments.
>
>  |        |              | Time in sec (mean (std)) | # Itr. (mean(std))
>   |---------- |----------|----------|---------|
> | n=10^4 | k=10^2 | 0.0011(0.00013)            | 6.2(0.4)
>  |n=10^5 | k=10^2 | 0.0108(0.00081)            | 8.0(0.2)
> | n=10^6 | k=10^2 | 0.13(0.0054)                   | 10(0)
> | n=10^7 | k=10^2 | 2.1(0.03)                          | 12(0)
> | n=10^8 | k=10^2 | 21.95(0.046)                   |13(0)
>
>
> Regarding the metric, we assume that we know the number of features in the ground truth, which is k=20 in the experimental setting. Furthermore, we made sure that all competing methods selected exactly k=20 features. We have elaborated this setup in the supplementary material between lines 389 and 391. Based on this setup, the ACC metric is measuring how many true supports are recovered out of the k=20 selected features, and therefore, a zero dominant vector will not influence the ACC score. We will clarify this in the paper in the next revision.
>
> Regarding the impact of k, we actually used arbitrary k in the experiments (line 202). To study the impact of k, we did an extra experiment. The results are shown in Table 2. We fix n but change k. As shown, the proposed projection algorithm has better convergence for large k.
>
> Table 2. Run time and #. Iterations on varying k over 100 experiments.
>
>  |               |             | Time in sec (mean (std))   | # Itr. (mean(std)) |
>   |---------- |----------|----------|---------|
>  |n=10^6 | k=10^2 | 0.13(0.0054)                   | 10(0)    |
>  |n=10^6 | k=10^3 | 0.12(0.0084)                   | 8.5(0.5)    |
>  |n=10^6 | k=10^4 | 0.11(0.0039)                   | 7(0)          |
>  |n=10^6 | k=10^5 | 0.10(0.0065)                   | 5.1(0.2) |
>
> We will include these two tables in the next version of the paper.

---

### Official Review · Reviewer_3a3C · 2021-07-17

**Rating:** 4
**Confidence:** 3

**Summary:**

In this paper the authors propose a method for projecting onto the capped simplex. The algorithm lies on a well-known framework, which transforms the optimization problem via Langrange and duality theory, in other equivalent problem. The contribution is on a different approach, using the Newton method, to solve one of the sub-problems. The algorithm is tested on a problem with genomic data.


**Limitations And Societal Impact:**

The authors claim that the bound of the input vector plays a role in the time-performance of their method, but more discussion is needed in that sense.

**Main Review:**

The contribution of the paper is the use of Newton's method for solving the subproblem (5). The main part of the paper is focused on the computation of derivatives and results related to them, many of them very basic. I think this part is not relevant, and in any case might be included in the supplementary material.

The modification of the method seems to be of importance, given the complexity discussed (even though it might not work fast for some vectors), but I think the presentation should be dramatically improved.

The texts is repetitive, the toy example in Fig.1 is not explained or motivated at all, and there are some technical parts with details or omissions. For instance, in line 26 it should say that the vector to be projected is $y$; problem (1) should be actually argmin; on the line 30, the convexity of $\Delta_k$ has to be used, otherwise the statement is false, Newton's algorithm doesn't have a stopping criteria, it runs for a fixed number $T$ of iterations, but $T$ is not an input, and is not discussed as it should.

Also, please proofread and check for typos, missing words, etc, and please be consistent with the notation, for instance in some parts $[]_+$ is used, and in other parts just $\max(0,\cdot)$, for the same argument.

I tend to think that the result is important, but unfortunately I think the paper needs more work.


**Time Spent Reviewing:**

3

---

> ### Author Response · Authors · 2021-08-10
> **Response to Reviewer 3a3C**
>
> We thank the reviewer for the comments. We agree that some paragraphs of the paper can be improved, and we thank the reviewer for pointing out these small mistakes. The typos will be fixed in our next version. However, we think the reviewer overlooked the contribution of the paper and we do not think that the comments on writing and small mistakes together give a fair justification that the paper has to be rejected. We now address each of the comments below.
>
>
>
> For the comment "in line 26 it should say that the vector to be projected is y; problem (1) should be actually argmin; Newton's algorithm doesn't have a stopping criteria, it runs for a fixed number T of iterations, but T is not an input, and is not discussed as it should."
>
> Reply: We will fix these in the next revision.
>
>
>
> For the comment "The main part of the paper is focused on the computation of derivatives and results related to them, many of them very basic. I think this part is not relevant, and in any case might be included in the supplementary material."
>
> Reply:  It is true that the Langrangian approach for solving projection onto simplex is well-known, but in this paper, the derivatives of the scalar minimization problem are not trivial nor basic at all (this is also pointed out by the 3rd reviewer). To the best of our knowledge, this is the first paper on using the Hessian of the scalar Langrangian multiplier variable to solve the projection problem. So we do not agree that the paper is basic. Furthermore, the paper seems to be basic because we tried our best to make the paper more accessible (which is illustrated by the comments of other reviewers that the paper is easy to read).
>
> For the comment "on the line 30, the convexity of Δk has to be used, otherwise the statement is false,"
>
>
> Reply: Continue from the last comment, our aim is to make the paper more accessible so we hide details in some parts of the paper and I think this causes misunderstanding. In fact, the reviewer’s comment is not accurate. Convexity (as pointed out by the reviewer) is actually not enough to ensure solution uniqueness, even for strong or strict convexity. Note that for some values of k, the capped simplex problem has no solution, and for some values of k the problem has many solutions, therefore the key to the solution uniqueness is not just on convexity but also on k (which we actually discussed in the discussion in the latter part of the paper).
>
> Here we also comment that the reviewer may have overlooked the contribution of the paper that, when the projection problem has no solution for some k, note that we stated the proposed method can still provide a solution for the projection problem when such a problem cannot be solved by the sorting-based approach.
>
>
> For the comment "the toy example in Fig.1 is not explained or motivated at all".
>
> Reply: Continue from the last comment, notice that Fig.1 is important as it showcases the effect of k on the solution of the projection problem (indicated by the interaction between lines l1 and l2 in the figure, if the two lines cross-over, then there is a solution), and we have already stated these (including the discussions on the previous comment) in the caption of figure 1. Furthermore, figure 1 is useful for providing a pictorial understanding of the ideas presented in the paper. In fact, In the later parts of the paper, several places can be redirected to figure 1 for a better understanding.
>
>
> For the comment "please proofread and check for typos, missing words, etc, and please be consistent with the notation, for instance in some parts []+ is used, and in other parts just max(0,⋅), for the same argument."
>
> Reply: We will fix these in the next revision.
>
> To sum up, we hope the reviewer can focus more on the technical contribution of the paper and reconsider it.

---

### Official Review · Reviewer_NzEp · 2021-07-20

**Rating:** 7
**Confidence:** 2

**Summary:**

This paper poses a new method for projecting vectors onto the k-capped simplex. This method reformulates the quadratic program into a form that can harness Newton method, a second-order iterative optimization algorithm, to solve the projection goal. The key novelty is using a second-order method to iteratively solve for a minimum, as opposed to sorting-based methods with a cubic complexity. Furthermore, first-order methods are shown to be too slow based on a proof that the Lipschitz constant scales with dimensionality. Compared to existing methods for k-capped simplex projection, the proposed method is several times faster, and in synthetic experiments achieves state-of-the-art accuracy. On experiments in high-dimensional bioinformatics, the proposed projection converges faster and equally well as other state-of-the-art methods.

**Limitations And Societal Impact:**

The authors have sufficiently addressed limitations and there are no clear societal impacts.

**Main Review:**

The method in this paper nicely extends the idea of capped simplex projection using Newton’s method. While the formulation in (5) appears elsewhere, the insight here is that approximate numerical methods can find a similarly high-quality solution to existing methods. Excluding typos, the writing throughout is clear, and the paper is well-organized and fairly easy to follow. The limitations are discussed in a fairly upfront manner, such as the bounded vector magnitude and lack of exact solution. The primary concern comes from a lack of applications that may be of interest to the community. The GWAS results are not analyzed for any biological significance, and while faster than previous methods, does not unlock any analysis that was not achievable prior. Furthermore, it would be helpful to have scaling results as a function of the number of samples, since the considered example is a very small GWAS. Modern GWAS are on the order of 100k’s of individuals. One such example of further impact would be to pick an appropriate sparsity level (k) based on cross-validation or other measures, since robustness and reproducibility are key. This might highlight the advantages of the proposed method better, and have further applications to hypothesis testing in GWAS.

Update to author response: please see discussion with authors regarding expanded experiments on synthetic GWAS data highlighting the effectiveness of the proposed method. Review score has been updated.

**Time Spent Reviewing:**

4

---

> ### Author Response · Authors · 2021-08-10
> **Response to Reviewer NzEp**
>
> First, we thank the reviewer for the comment. In summary, the main focus of the paper is to develop a fast and efficient projection algorithm for the capped simplex projection. The main focus of section 4 is to showcase the projection algorithm that can be used to solve the well-established “sparse Least-square regression” model ($P_{BR}$) [1,2,3] efficiently. All the concerns raised by the reviewer are related to ($P_{BR}$) itself, which are in fact not the focus of the paper. Therefore, we hope that the reviewer could reconsider the paper and the rating score. In the following, we address each comment in detail.
>
> Review comment 1: “The primary concern comes from a lack of applications that may be of interest to the community.”
>
> Reply: The application of the proposed projection algorithm is to solve the “sparse Least-square regression” described in ($P_{BR}$) in section 4. The problem ($P_{BR}$) is a well-established model, which has drawn great attention in the statistical community last year [1,2]. It has been demonstrated that it can significantly outperform traditional sparse regression methods (e.g. lasso and elastic net) when the sample size is small and the feature dimension is huge.
>
> In section 4, we elaborated on how we can use the proposed projection algorithm to solve ($P_{BR}$). We empirically demonstrated that the algorithm using the proposed projection method outperforms all current methods in terms of efficiency (computational time). Detailed results are shown in Fig. 3 and Fig. 4 in the paper and also Fig. 5 in the supplementary document.
>
> To sum up, we have shown an important application of the proposed projection algorithm to solve the “sparse Least-square regression” model ($P_{BR}$) [1,2,3] in section 4.
>
>
> Review comment 2: “The GWAS results are not analyzed for any biological significance, and while faster than previous methods, does not unlock any analysis that was not achievable prior.”
>
> Reply: We want to emphasize that the goal of the GWAS experiments is to show that the algorithm that uses the proposed projection method has a better convergence rate than existing algorithms on real-world data. The two methods we compared in the GWAS experiments aim to solve the same “sparse Least-square regression”  model ($P_{BR}$). As shown in Fig. 4, the algorithm using the proposed projection algorithm has better convergence.
>
> Because both algorithms we compared in Fig. 4 are solving the same problem ($P_{BR}$). They have the same solution (when converging), which is true as they converge to the same objective function value. Therefore, there is no difference between their results in terms of biological significance.
>
>
>
>
> Review comment 3: “Furthermore, it would be helpful to have scaling results as a function of the number of samples, since the considered example is a very small GWAS. Modern GWAS are on the order of 100k’s of individuals.”
>
> Reply: We picked the GWAS data with a small sample size on purpose. As illustrated in Fig. 3 and [1,2,3], the model ($P_{BR}$) outperforms traditional sparse regression models when the sample size is very small and the feature dimension is large. We have stated the strength of ($P_{BR}$) in the paper between line 220 and line 225. Therefore, in order to showcase the strength of the model ($P_{BR}$), both the simulation data and the real-world GWAS data used in the experiments in section 4 need to have the property that the sample size is much smaller than the feature dimension. When we have enough samples (such as GWAS data of 100k’ of individuals), traditional methods are recommended [1,2], but this is out of the scope of the paper.
>
>
> Review comment 4: “One such example of further impact would be to pick an appropriate sparsity level (k) based on cross-validation or other measures, since robustness and reproducibility are key.”
>
> Reply:  The problem of ``how to select k’’ has already been investigated in [2] and this is not the focus of the paper.
>
>
>
> References
>
> [1] D. Bertsimas and B. V. Parys. Sparse high-dimensional regression: Exact scalable algorithms and phase transitions. The Annals of Statistics, 48(1):300 – 323, 2020.
>
> [2] D. Bertsimas, J. Pauphilet, and B. V. Parys. Sparse Regression: Scalable Algorithms and Empirical Performance. Statistical Science, 35(4):555 – 578, 2020.
>
> [3] M. Pilanci, M. J. Wainwright, and L. El Ghaoui. Sparse learning via boolean relaxations.
> Mathematical Programming, 151(1):63–87, 2015.

---

> > ### Comment · Reviewer_NzEp · 2021-08-31
> > **thank you**
> >
> > Thank you for the detailed responses and follow up.
> >
> > The responses to other reviewers to elucidate the relationship between T and n has been helpful and fills in some missing gaps in the analysis.
> >
> > The present review, strengthened by the author response, appreciates the novelty of the newly proposed method as well as its empirical performance. As other reviewers have mentioned and agree upon. It is clear the paper is not solely focused on solving $P_{BR}$, however this is the motivating example of the real results. It would simply be helpful to expand upon the attention given to solving $P_{BR}$ and the practical importance there, particularly if there are motivating examples and applications in [1,2] that help strengthen this case. That would be enough to push the review over the acceptance criteria. At the moment, it is still unclear the implications of the increased efficiency.

---

> > > ### Author Response · Authors · 2021-09-01
> > > **Respond to Reviewer NzEp**
> > >
> > > We thank the reviewer for the feedback. In fact, we tried to compare different methods for identifying the significant SNPs in the GWAS data used in the main text after the reviewer’s first feedback. However, since we do not know the ground truth, hence it is hard for us to evaluate the performance of different methods, and therefore we do not have a comparison between the method in the main text. In fact, we compared different methods using the simulated GWAS data described in a well-cited paper [1] (see below) after we submitted our author response. Below we describe the experiments we performed and we hope that these can answer the question from the reviewer.
> > >
> > > We compared $P_{BR}$ with Lasso and Elastic Net (ENet). We used the out-of-sample MSE to select the parameter k for $P_{BR}$. For Lasso and ENet, we used cross-validation with the minimum MES+1SE to tune the hyper-parameters, which was suggested by [1]. There are three different simulated GWAS datasets as stated in [1], in which they have different levels of correlation (LD: High, Mixed, and Low) between SNPs. All the datasets have 50,000 SNPs. We varied the number of sample sizes and showed the comparison results in Tables 1-4 below. Here we report the median and the standard deviation (in parentheses) over 100 replicates. We can see that $P_{BR}$ can find more correct SNPs with fewer false positives, especially when the sample size is small (in which we showed a similar result in Fig.3 in the paper, see the part ``Sparse Regression on Simulation Data’’). We believe these extra experiments help demonstrate the practical impact of solving $P_{BR}$ using the proposed algorithm in the main text and hopefully answer the question raised by the reviewer.
> > >
> > > In the next version, we will put Table 1 and Table 3 in the main text and put Table 2 and Table 4 in the supplementary materials.
> > >
> > > Table 1. Sample size=1,000, SNP=50,000, significant SNPs = 25
> > >
> > > |         |        |Lasso | ENet | $P_{BR}$ |
> > > |---------|--------|---------|--------|-----|
> > > |High LD  |Correct |3(0.91)  |25(0)   |25(0)|
> > > |         |False positive|0(0)|0(0.1) |0(0) |
> > > |Mixed LD  |Correct |3(0.75)  |18(1.47)   |25(0)|
> > > |         |False positive|0(0)|0(1.19) |0(0) |
> > > |Low LD  |Correct |17(2.12)  |25(0.51)   |25(0)|
> > > |         |False positive|0(0.39)|0(2.06) |0(0) |
> > >
> > > Table 2. Sample size=150, SNP=50,000, significant SNPs = 25
> > >
> > > |         |        |Lasso | ENet | $P_{BR}$ |
> > > |---------|--------|---------|--------|-----|
> > > |High LD  |Correct |2(1.91)  |24(1.37)   |25(0.46)|
> > > |         |False positive|1(0.92)|10(1.92) |1(0.47) |
> > > |Mixed LD  |Correct |3(1.75)  |19(1.87)   |24(0.87)|
> > > |         |False positive|1(0.63)|16(1.09) |1(0.24) |
> > > |Low LD  |Correct |7(1.23)  |20(0.81)   |25(1.26)|
> > > |              | False positive|3(0.72)|18(1.26) |1(0.27) |
> > >
> > > Table 3. Sample size=100, SNP=50,000, significant SNPs = 25
> > >
> > > |         |        |Lasso | ENet | $P_{BR}$ |
> > > |---------|--------|---------|--------|-----|
> > > |High LD  |Correct |2(1.01)  |20(3.47)   |25(1.52)|
> > > |         |False positive|1(0.93)|4(1.41) |1(1.03) |
> > > |Mixed LD  |Correct |2(1.35)  |16(1.47)   |24(2.03)|
> > > |         |False positive|1(0.68)| 2(0.35)|2(1.24) |
> > > |Low LD  |Correct |5(2.12)  |17(3.51)   |23(3.07)|
> > > |         |False positive|2(0.59)|5(2.06) |3(1.47) |
> > >
> > > Table 4. Sample size=50, SNP=50,000, significant SNPs = 25
> > >
> > > |         |        |Lasso | ENet | $P_{BR}$ |
> > > |---------|--------|---------|--------|-----|
> > > |High LD  |Correct |2(1.21)  |17(4.47)   |19(3.52)|
> > > |         |False positive|4(1.93)|14(3.41) |5(2.03) |
> > > |Mixed LD  |Correct |2(1.35)  |12(2.47)   |16(2.23)|
> > > |         |False positive|2(1.68)| 8(2.35)|6(2.24) |
> > > |Low LD  |Correct |1(0.32)  |12(3.61)   |15(3.27)|
> > > |         |False positive|0(0.19)|31(5.06) |10(3.47) |
> > >
> > > [1] Waldmann, P, Mészáros, G, Gredler, B, Fuerst, C, Sölkner, J (2013). Evaluation of the lasso and the elastic net in genome-wide association studies. Front Genet, 4:270.

---

> > > > ### Comment · Reviewer_NzEp · 2021-09-01
> > > > **Simulated GWAS data experiments**
> > > >
> > > > Thank you for the follow up experimental results and for elucidating the application. These results are quite convincing that the proposed objective function and solution can have impactful results for significance testing in GWAS. These experiments nearly completely address the concern of application impact, and are enough to change the review score. Fig 4 still shows Gurobi is feasible and converges in a reasonable time, however the proposed method does improve significantly upon that convergence.
> > > >
> > > > Noting that there is the issue of population structure correction, as addressed by [1], but that this is out of scope of the current work.

---

> > > > > ### Author Response · Authors · 2021-09-01
> > > > > **Thank you for your support**
> > > > >
> > > > > We thank the reviewer for helping us improve the paper. We hope the reviewer could kindly change the review score.

---

### Decision · Program_Chairs · 2021-09-27

**Decision:**

Accept (Poster)

**Comment:**

This paper poses a new method for projecting vectors onto the k-capped simplex. This method reformulates the quadratic program into a form that can harness Newton method, a second-order iterative optimization algorithm, to solve the projection goal. The key novelty is using a second-order method to iteratively solve for a minimum, as opposed to sorting-based methods with a cubic complexity.

Out of the 4 reviewers, 3 support accepting the manuscript. Detailed technical comments are provided in the reviews and the authors responded to these reviews in detail and appropriately. One reviewer suggests to reject the paper based on the fact that the manuscript has typos and an opinion that some elements of the manuscript would be better moved to the supplement. In essence, I think the low score by that reviewer is not well justified and in my own assessment of this manuscript I have excluded it. What's remaining are three reviewers that support the paper quite strongly overall (scores 7,7,6). I therefore recommend to accept this manuscript.